# Intra-Laboratory Evaluation of DNA Extraction Methods and Assessment of a Droplet Digital PCR for the Detection of *Xanthomonas* *citri* pv. *citri* on Different *Citrus* Species

**DOI:** 10.3390/ijms23094975

**Published:** 2022-04-29

**Authors:** Nicoletta Pucci, Valeria Scala, Giuseppe Tatulli, Alessia L’Aurora, Simone Lucchesi, Manuel Salustri, Stefania Loreti

**Affiliations:** 1Council for Agricultural Research and Economics—Research Centre for Plant Protection and Certification of Rome, Via C. G. Bertero 22, 00156 Rome, Italy; nicoletta.pucci@crea.gov.it (N.P.); valeria.scala@crea.gov.it (V.S.); giuseppe.tatulli@crea.gov.it (G.T.); alessia.laurora@crea.gov.it (A.L.); simone.lucchesi@crea.gov.it (S.L.); 2Department of Environmental Biology, University of Rome “Sapienza”, 00185 Rome, Italy; manuel.salustri@uniroma1.it

**Keywords:** citrus bacterial canker, *Citrus lemon*, *Citrus sinensis*, diagnostics, DNA quality

## Abstract

*Xanthomonas citri* pv. *citri* (Xcc) and *X. citri* pv. *aurantifolii* (Xca), causal agents of citrus bacterial canker, are both regulated by the European Union to prevent their introduction. Xcc is responsible for severe outbreaks of citrus production worldwide, therefore, a prompt and reliable detection is advisable for the early detection of this bacterium either in symptomatic or asymptomatic plant material. The current EPPO (European and Mediterranean Plant Protection Organization) diagnostic protocol, PM 7/44(1), includes several diagnostic tests even if new assays have been developed in the latter years for which validation data are needed. Recently, a test performance study was organized within the Valitest EU Project to validate Xcc diagnostic methods and provide evidence on the most reliable assays; however, the influence of DNA extraction methods (DEM) on the reliability of the detection has never been assessed. In this study we evaluate four different DEM, by following two different approaches: (i) a comparison by real-time PCR standard curves of bacterial DNA versus bacterial DNA added to plant DNA (lemon, leaves and fruit; orange fruit); and (ii) the evaluation of performance criteria of spiked samples (plant extract added with ten-fold diluted bacterial suspensions at known concentrations). Droplet digital PCR is developed and compared with real-time PCR, as the detection method.

## 1. Introduction

Citrus bacterial canker (CBC), caused by *Xanthomonas citri* pv. *citri* (Xcc) and *X. citri* pv. *aurantifolii* (Xca), is one of the most serious diseases of several host species of the Rutaceae family, among which is included *Citrus*, Fortunella, *Poncirus* and their hybrids [1]. Xcc and Xca are the causal agents of the “Asiatic citrus canker” and the “South American citrus canker” (or sometimes false canker), respectively, and the symptoms they cause are morphologically indistinguishable [2]. CBC is mostly a leaf- and fruit-spotting disease, but when conditions are highly favorable, infections cause defoliation, shoot dieback and fruit drop. The two pathovars are distinguished in pathotypes, Xcc in pathotypes A, A^w^, and A* and Xca in pathotypes B and C. Pathotype A has a wide host range of *Citrus* and other related genera and a worldwide distribution causing severe outbreaks. Pathotypes A^w^ and A* have a narrow host range and are limited, respectively, in Florida (USA) and Asia. Pathotypes B and C both infect *C. aurantifolia,* have a restricted host-range and are reported in South America (Argentina, Brazil, Paraguay and Uruguay), and no significant outbreaks have been reported in recent years, suggesting that it has been outcompeted [3]. Both pathovars are included in the Annex II, part A of Regulation (EU) 2019/2072 [4]: this implies that their introduction and spread is banned within the EU member states.

Moreover, Annex IV, part A, Section 1, of the Directive 2000/29/EC [5] laid down special requirements for the fruits of several *Rutaceous* genera to ensure that imported commodities are free from the pest. Risks factors for the introduction of the disease are related to plant propagating material (trade, movement, import and preparation), but also the handling and processing of imported infected fruit near to *Citrus* orchards [6]. Natural (splashing, aerosol, wind-driven rain) and human dispersal (movement of infected plant material, fruit included, machinery, clothes, tools used in agricultural practice, citrus in urban areas) also contributes to the dissemination and spread [6]. An early detection based on surveillance activity (performed in the period of flushing and leaf expansion of the plant, or after adverse weather events) is crucial to avoid accidental entry of the pathogens. Moreover, in consideration of latent infections occurrence, the detection of the pathogens needs to be focused either on symptomatic or on asymptomatic plant material [6]. The tests available for the diagnosis of Xcc and Xca are reported in the European and Mediterranean Plant Protection Organization (EPPO) PM 7/44(1) [1] and in the International Standards for Phytosanitary Measures (ISPM) 27 [7]. New molecular detection methods have been recently described, but no validation data were available on the above-mentioned official protocols. In the frame of the Valitest EU Project (GA n°773139) a test performance study (TPS) was organized for the validation of several diagnostic methods aimed at detecting Xcc. Otherwise, no information about the possible influence of DNA extraction methods (hereafter DEM) is available. It is well known that the DNA extraction step could be very crucial for the reliability of the molecular test. DNA polymerase inhibitors, such as polysaccharides and phenolic compounds, need to be removed to avoid the occurrence of false-negative results in the detection assay [8,9].

The purpose of this intra-laboratory study was to compare several DEMs to assess their possible influence on the detection of Xcc in association to several plant matrices. This evaluation was performed following these approaches: (i) a comparison by real-time PCR, Cubero and Graham (2005) [10], of the standard curves of bacterial DNA versus bacterial DNA added with plant DNA (spiked DNA); (ii) the evaluation of performance criteria of plant extract added with ten-fold diluted Xcc bacterial suspensions at known concentrations (spiked extract); and (iii) evaluation of the results obtained by droplet digital PCR (ddPCR) adapted from Cubero and Graham (2005) [10]. Four different DEM were selected, among the most used, in laboratories performing analyses of bacterial pathogens: DNeasy Plant Mini kit Qiagen (Plant), DNeasy Mericon Food kit Qiagen (Mericon Food), CTAB-based method (Ctab), and QuickPick™ SML Plant DNA kit, Bionobile (Quick). The host plants lemon and orange were selected as represented the most widespread citrus crops in Mediterranean countries, which are the areas with the main risk of introducing the disease. For this study a representative strain of Xcc pathotype A was used, since it has a broader host-range and wider diffusion in the territory.

## 2. Results

### 2.1. Samples Homogeneity

The evaluation of homogeneity was performed for samples SET1 and SET2 (see material methods for details). The lower DNA (1 fg/reaction, SET1) and bacterial (10^2–10^ cfu/mL, SET2) concentrations showed a high SD (Standard Deviation) (>1), consequently, these samples were not further considered for the standard curve evaluation (SET1) or for the evaluation of performance criteria (SET2), respectively. In particular, all samples at 1 fg/reaction and 10 cfu/mL, and some samples at 10^2^ cfu/mL, showed a high incidence of poorly homogeneous samples and/or Ct values very close to those recorded for the previous dilution (ΔCt was less than three cycles as expected).

### 2.2. Development of Droplet Digital PCR

The optimal annealing/elongation temperature was identified at 58 °C, either with 5 or 8 µL of DNA per reaction. At this temperature, the three following parameters were satisfied: (i) positive droplets with the highest fluorescence amplitude; (ii) a better separation between positive/negative droplets; and (iii) less rain (i.e., droplets ranging between the positive and negative ones) (data not shown). The optimal selected volume was 8 µL, as the increase in DNA volume resulted in a higher number of positive droplets with respect to 5 µL. The detection was successful testing the SET1 and SET2 samples, for all DEM/plant matrices combinations (Table 1 and Table 2) and samples of bacterial DNA from strain Xcc NCPPB 3234 (Table 3). Figure 1 shows the results obtained on samples of SET2 from 10^3^ to 10 cfu/mL.

### 2.3. Evaluation of SET1 Samples

The raw data of real-time PCR and ddPCR results for SET1 are reported in Table 1, according to the DEM and the different plant matrices. The Table 3 shows the result of the standard curve obtained by real-time PCR on bacterial DNA from a pure culture. The amplification efficiency of DNA from the Xcc pure culture showed an adequate value (E = 98.17%); this was comparable to the efficiency shown by most of the combinations of DEM/plant matrices of SET1, which give acceptable values between 90–105%. The exceptions were orange fruit/Mericon (E = 88%), orange fruit/Quick (E = 89%), and lemon leaf/Ctab (E = 111%), whose E values slightly deviate from the acceptable values.

The comparison between the standard curves of bacterial DNA and SET1 (Figure 2), highlight an overlapping for all plant species/matrices, that was particularly evident at the higher bacterial concentration. On the other hand, at lower bacterial concentration some differences were observed due to a higher difference among the Ct (in particular at 10 fg/real-time PCR reaction) (Figure 2). This was predictable at low target concentrations due to the greater variability between technical replicates.

The range of Ct values for SET1 samples and for bacterial DNA are graphically represented in Figure 3. The range of Ct were not particularly variable among the different combinations of DEM and plant matrices, showing comparable results with bacterial DNA.

The Bland–Altman plot reported in Figure 4 shows the percentage variation of the average of sample concentrations of SET1 (represented by the black line of Figure 4) with respect to a null variation (represented by the red line in Figure 4). The percentage variation indicates whether the method underestimates or overestimates the overall Ct values. A good performance of the method implies that the black line approaches the red one, indicating a low variability. All DEM/plant matrices combinations were within the 95% of confidence interval included in the graph by black dashed lines. In particular, the Plant DEM gave the best performance with all plant matrices, except for the orange fruit.

In order to assess the effect of inhibitors related to the different DEM/plant matrices combination, the SET1 was analyzed by ddPCR, a technique known to be less affected by inhibitors with respect to real-time PCR [11]. The results showed that at the lowest DNA target concentration (10 fg/ddPCR), the orange fruit matrix resulted in the highest number of negative results; however, among the DEM, Plant showed reliable results with all plant matrices as shown in Table 1. Quick showed accurate results with the exception for orange fruit, Ctab was reliable only in samples of lemon leaf and Mericon in none (Table 1).

### 2.4. Evaluation of Healthy Samples and Performance Criteria of SET2

The analysis by real-time PCR [10] of 20 healthy randomly selected samples, sporadically yielded high Ct values and/or inconsistencies between the two technical replicates (i.e., high Ct value/NA) (Appendix A). As reported in Table 4 high Ct/inconsistencies were in different percentages depending on the plant matrices or DEM. In particular, the higher percentages occurred by using the Ctab method (43%), with respect to the Mericon and Plant (respectively, 28% and 25%), with lower values for the Quick (11 %). The healthy status of these 20 samples was assessed by droplet digital PCR, giving a negative result (data not shown).

The results of the real-time PCR and ddPCR on SET2 samples are reported in Table 2. The homogeneity showed an acceptable range of variation within bacterial concentrations from 10^7^ to 10^3^ cfu/mL of SET2 samples (Table 2). The results of performance criteria (Table 5) showed 100% of diagnostic specificity, diagnostic sensitivity, and accuracy for all DEM/ plant matrices combinations.

The analytical sensitivity (ASE) was evaluated on SET2 considering different DEM/plant matrices. In particular, ASE was estimated for real-time PCR [10] and the ddPCR in a range of 10^2^–10^3^ cfu/mL and 10–10^2^ cfu/mL, respectively, depending on the method used for the DNA extraction (Table 6). By real-time PCR, Quick and Plant showed better results, than Ctab and Mericon. The ASE of 10^2^ cfu/mL was obtained by Quick and Plant except for one plant matrix (lemon fruit and orange fruit, respectively) which gave 10^3^ cfu/mL. Using Ctab, the ASE was 10^2^ cfu/mL only for one out of three plant matrices (lemon leaf) and 10^3^ cfu/mL for the other matrices, whereas for Mericon the ASE was at 10^3^ cfu/mL for all plant matrices. By ddPCR, using Plant, Mericon and Ctab, the ASE was 10 cfu/mL for two out of three matrices (10^2^ cfu/mL was obtained for lemon fruit and orange fruit depending on the DEM) whereas by Quick the ASE was 10 cfu/mL for all plant matrices (Table 6).

## 3. Discussion

The diagnosis of plant pathogenic bacteria represents one of the more important challenges aimed to prevent the dissemination of world-wide diseases. The adoption of a reliable detection method is crucial to correctly determine the presence of a pest before its establishment in a pest-free area and/or its spread. To standardize the use of diagnostics, in plant pathology it is suggested that their validation follow UNI EN ISO 16140 [12], ISO/IEC 17025 [13] and EPPO protocols that allow to evaluate the performance of tests and/or laboratories [14,15]. Intra- or inter-laboratory studies permit the evaluation of several parameters for each diagnostic test in terms of sensitivity, specificity, repeatability, and reproducibility; however, a crucial step is the preparation of DNA extracts that could interfere with the ability of a test to detect the target, regardless of its performance (e.g., sensitivity, specificity).

Furthermore, the EPPO protocols for diagnostics generally do not require DNA quality control as a mandatory step, therefore the DNA is not subjected to control before performing a test. Neither of the majority of diagnostic methods for phytopathogenic bacteria foresee the use of internal DNA amplification controls in each sample (i.e., the mitochondrial cytochrome oxidase, *cox* gene; the region coding for the 5.8S rRNA, 5.8S rDNA gene), rather, additional samples (PIC, positive isolation) are provided to guarantee the success of the DNA extraction. Moreover, many commercial kits based on standardized procedures have been developed which can ensure the successful outcome of the analysis, and today many kits based on different principles or systems (i.e., columns, beads, manual, automatized) are available. However, the choice of the most appropriate extraction kit is related to various factors ranging from technical aspects (e.g., type of plant matrix) to the economic availability of the laboratory. Taking this last aspect into account, still today many designated laboratories that carry out official analyses use home-made DEM based on the use of Ctab, as for the detection of *Xylella fastidiosa* [16], one of the most alerted phytopathogenic bacteria in Europe and worldwide.

Xcc, along with Xca, are listed by the EU as a priority pest. Both pathogens were included in the EURL-BAC work program in 2019–2020 with the aim of standardizing, inventorying test protocols, developing and validating detection tests, and providing reference material to standardize detection/identification and facilitate disclosure to national reference laboratories (NRLs). In the case of Xcc, a TPS validation was recently organized within the Valitest Project (XCC-1) (GA n°773139), aimed at validating several diagnostic methods for Xcc detection. The obtained results provided useful indications on the validated diagnostic methods, highlighting their performance. An adequate availability of diagnostic tools for the detection of Xcc emerged from the report by the Valitest Project (XCC-1) [17], highlighting that the real-time PCR techniques are more sensitive than respective conventional PCR, except for the real-time PCR of Mavrodieva et al. [18]. This validation activity did not consider the DNA extraction and the influence that this step may have for the reliability of the detection both in conditions of high and low levels of infection of the plant sample, or in relation to different plant matrices. To avoid the occurrence of false negative results in the molecular detection test, the DNA extraction step must ensure the effective removal of DNA polymerase inhibitors [8,9]. In this study, the possible influence of several combinations of DEM/plant matrices were assessed with the real-time PCR detection test of Cubero and Graham [11]. The real-time PCR developed by Ròbene et al. [19] was not taken into consideration as the primers/probe had not yet been published at the time of the TPS execution. The Valitest TPS has been considered reliable for the detection of Xcc for both tests [17]. For this study, Xcc pathotype A was selected, considering its broader plant host-range and wider spread in the infected areas, compared to the A^W^ and A* pathotypes [5]. The overall results showed that at high-medium bacterial (10^7^–10^3^ cfu/mL) or DNA (1 ng-100 fg/PCR reaction) concentrations all DEM allow a reliable detection of Xcc from the tested plant matrices (lemon and orange fruits and lemon leaves). This evidence was highlighted: (i) by the overlapping of the standard curves of bacterial DNA and the samples of SET1; (ii) by the acceptable efficiency values for most samples of SET1 showing a scarce influence of the DNA quality on the amplification efficiency; and (iii) by the value of 100% of diagnostic sensitivity, diagnostic specificity and of the accuracy for samples of SET2.

At the lowest concentrations (10 fg/PCR reaction-SET1 and 10 cfu/mL-SET2), most of the samples did not show an acceptable homogeneity or a reliable result by real-time PCR, which resulted in variable results, frequently non-consistent, depending on the plant matrices /methods combination. This evidence shows that at this condition the DNA extraction methodology can affect the reliability of the detection tests. The evaluation of the ASE of SET2 (spiked samples) by the real-time PCR of Cubero and Graham [10] showed a range of 10^2^–10^3^ cfu/mL, in accordance with the results of the XCC-1 TPS Valitest [17]; however, this value ranged depending on the different DEM/plant matrices combination, showing the best ASE values for Quick and Plant, intermediate for Ctab and the worst for Mericon.

The ddPCR is notoriously less affected by inhibitors [11]. The presence of inhibitors has the potential to increase errors, reducing the method resolution and producing false negative results [20]. Our ddPCR outcome showed negative results with most of the samples of the Xcc SET1 at 1fg/ddPCR due to the limit of detection of the method. At 10fg/ddPCR, negative results were obtained by Mericon with all matrices, by Ctab with orange and lemon fruit, by Quick with orange fruit, whereas Plant detected the target by all matrices. By assessing SET2, the ddPCR showed a ten-fold higher ASE with respect to real-time PCR, with the best ASE values obtained when Quick was used for the DEM.

The overall results indicate that the use of commercial kits is adequate for a reliable DNA extraction for all the tested plant matrices at a high concentration of the target. Particular attention must be paid to the choice of the most appropriate extraction method when processing samples with an expected low bacterial load, as in the case of asymptomatic samples. In this case, the use of Quick and Plant seems the most reliable DEM. An adequate choice of the DNA extraction procedure allows to improve the sensitivity of the method by one decimal factor. In this regard, the adaptation of a ddPCR system according to the Cubero and Graham test [10] has shown that the latter is more sensitive than the respective real-time PCR method. Dupas et al. (2019) [21] reported that the sensitivity of ddPCR compared to real-time PCR is controversial, because some authors have reported an increasing in sensitivity but others showed a 10–100-fold lower sensitivity. The benefits of ddPCR seem to be related to the studied pathosystem [22], e.g., the detection of *Ralstonia solanacearum* was improved using ddPCR compared to real-time PCR but was comparable for *Erwinia amylovora*. The ddPCR adapted from Harper et al. (2010) [23] for the detection of *Xylella fastidiosa* showed the same sensitivity of the corresponding real-time PCR for several plant species (*Olea europaea, Polygala myrtifolia* and *Rosmarinus officinalis*) and an improved sensitivity for others (*Quercus ilex* and *Lavandula angustifolia*) [21], that are known to be rich in plant inhibitors. An improved sensitivity of the ddPCR may therefore depend both on the higher tolerance of the ddPCR to inhibitors and on the lower coefficient of variation of the ddPCR compared to real-time PCR, especially at low target concentrations [12]. Zhao et al. [11] developed a ddPCR for the detection of Xcc reporting its potential for quantitative detection with a high precision and accuracy with respect to a real-time PCR assay, and its suitability for the diagnosis of pathogens in field samples with a complex matrix of inhibitors.

Finally, in our own laboratory conditions late Ct values occurred by real-time PCR on healthy plant matrices, with a high incidence for Ctab and low for Quick, supporting the argument that Quickis the most suitable method for DNA extraction. A ddPCR was performed to evaluate if the late Ct were due to background contaminations and the obtained negative results confirmed that these late Ct values were false signals. We suggest establishing a cut-off for real-time PCR [10] to avoid the occurrence of false-positive, due to late Ct values, that interfere with the detection of the bacterium. This aspect must be considered particularly for the analysis of asymptomatic material, where the choice of an adequate DEM could be crucial for the success of the diagnosis. In conclusion, this intra-laboratory study highlighted that all DEM can be used for the analysis of symptomatic material; however, the choice of Quickand Plant seem the most appropriate for the successful detection of Xcc in samples with a low concentration of the pathogen, such as in asymptomatic material. It is worth noting that the surveillance protocol should consider possible asymptomatic but infected material (EFSA) [6]. Indeed, it must be taken into account that the major risk factors for the long-distance spread capacity of Xcc are represented by the commercial shipment of diseased or contaminated fruit [24] or plant propagating material (e.g., budwood, rootstock, seedlings) [2,25].

Although molecular diagnostic tests allow for reaching the levels of sensitivity adequate for the detection of Xcc in low concentrations, the validation data produced in this study show that the wrong choice of the extraction method can affect the outcome of official analyses, especially in critical samples (i.e., low concentration of the pathogen, pest-free areas). Conversely, applying reliable diagnostics will help protect pest-free areas from accidental introduction of this quarantine pathogen.

## 4. Materials and Methods

### 4.1. Bacterial Strain

The strain NCPPB 3234 (Xcc) was used for the samples’ preparation. This strain, belonging to the National Collection of Plant Pathogenic Bacteria (NCPPB Fera Science Ltd - UK), was isolated in 1982 in Japan from *Citrus* spp. and belong to pathotype A. The lyophilized strain was grown in NGA (nutrient agar added with 0.25% d-glucose) at 28 °C. Bacterial suspensions were prepared in phosphate saline buffer (PBS 10 mM, pH = 7.2) and spectrophotometrically (DS-11 Fx+, Spectrophotometer-Fluorometer Denovix Inc., Wilmington, DE, USA) measured at a concentration corresponding to about 10^8^ colony forming units (CFU) mL^−1^ (OD_660_ = 0.1). The ten-fold dilution of bacterial DNA was prepared as a standard reference curve.

### 4.2. Plant Material

Plant material of the host plants was collected from plants in open fields (leaves) or bought in trade markets (fruits) located in the Latium and Campania Regions (Italy) (Table 7).

The plant extract was prepared as reported in [6] with some modifications as indicated below. Thirty grams of leaf tissue and/or fruit surface layer was collected for the orange and lemon (Table 7) and cut into smaller pieces. Each sample was subdivided into 3 subsamples of 10 g, washed with PBS in sterile bags (Bioreba, Reinach, Switzerland), and shaken for 20 min at room temperature. After incubation, the supernatant was filtered to remove the plant material and centrifuged for 20 min at 10,000× *g*. The pellet of each subsample was resuspended in 10 mL of PBS. Finally, the three subsamples were combined into one sample to proceed with the steps reported below. The healthy status of the plant extract was assessed for each plant matrix by real-time PCR [10] before proceeding with samples preparation and used only in the case of negative results. The plant extracts were used for the plant DNA extraction performed with four different DEM as reported below.

### 4.3. Sample Preparation

Samples prepared for the intra-laboratory study were grouped in SET1 and SET2. SET1 was prepared by amending the DNA of the plant extract (orange fruit, lemon leaves and lemon fruit) obtained with the four different DEM, with a ten-fold dilution of Xcc DNA (from 1 ng to 1 fg/real-time PCR or ddPCR reaction). SET2 consisted of spiked samples prepared by adding bacterial suspensions at known concentrations (CFU/mL) to healthy plant extracts (orange fruit, lemon leaves and lemon fruit). The obtained spiked samples (bacterial suspension + plant extract) were extracted by DEM. The bacterial suspension was added in the different plant matrices at a final concentration of Xcc 10^7^, 10^6^, 10^5^, 10^4^, 10^3^, 10^2^, and 10 cfu/mL. Moreover, a plant extract without the addition of bacteria was prepared (NIC). Three independent biological replicates (for each plant matrix, extracted with the different DEM) were prepared for each bacterial concentration. The number of colonies forming units (CFUs) was determined by plating 100 µL of bacterial suspensions on an NGA medium and incubating at about 27 °C and determined after 4 days. Twenty samples for each plant matrix randomly selected were extracted with the four DEM and evaluated by real-time PCR and droplet digital PCR as non-target samples for the evaluation of the performance criteria.

### 4.4. DNA Extraction

Total bacterial DNA was extracted from 1 mL of bacterial cultures of Xcc NCPPB 3234 using Gentra Puregene Yeast/Bact. Kit (Qiagen, Venlo, The Netherlands). The DNA concentration was evaluated by Qubit (dsDNA HS Assay kit, Invitrogen, Waltham, MA, USA). Plant DNA extraction was performed following the manufacturers’ instructions of the commercial kits, i.e., DNeasy Plant Mini kit (Qiagen, Venlo, The Netherlands), DNeasy Mericon TM Food Kit (Qiagen, Venlo, The Netherlands), and QuickPick™ SML Plant DNA kit (QRET Technologies Ltd., Turku, Finland), manual version, from 500 mL of plant extract. The Ctab DEM was performed as described for *Xylella fastidiosa* [16]. The DNA was stored ≤−15 °C until analysis.

### 4.5. Real-Time (Real-Time PCR)

The Cubero and Graham [11] real-time PCR reaction was performed according to the TPS instructions for the detection of Xcc (TPS code: XCC1) [17] received in the frame of the EU project Valitest (GA n°773139). TaqMan reactions were carried out in a 25 μL reaction mixture containing 0.4 mM of each primer (J-pth3/4), (Eurofins Genomics Germany GmbH, Ebersberg, Germany) 1× GoTaq® Probe qPCR Master Mix (Promega Corporation, Madison, WI, USA), 0.2µM TaqMan probe J-Taqpth2 (Eurofins), 5 µL of DNA extracts; the thermal profile consisted of 45 cycles at 95 °C-10 min, 95 °C-15 s, and 60 °C 1 min. All real-time PCR runs were performed including at least a control sample of PAC (positive amplification control), NIC (negative internal control) and NAC (negative amplification control). The samples of SET1 were run in one biological replicate, each in three technical replicates (*n* = 3). The samples of SET 2 were run in three biological replicates (for each plant matrix), each in technical triplicate (*n* = 9).

### 4.6. Droplet Digital PCR (ddPCR)

The ddPCR was developed adapting the method of Cubero and Graham [10]. For this purpose, the optimal annealing/extension temperature was identified by observing three different parameters: (i) the highest fluorescence amplitude; (ii) the positive/negative droplets disengagement; and (iii) less rain (i.e., droplets ranging between the positive and negative ones).

Following Biorad ddPCR^TM^ Supermix datasheet information, two thermal gradients were tested to determine the optimal hybridization temperature ranging from 53 °C to 60 °C on 5 µL or 8 µL of DNA Xcc, strain NCPPB 3234. The optimized ddPCR reaction mix retained consisted of 1xddPCRTM Supermix for Probes no dUTP ( Bio-Rad Laboratories Inc., Hercules, CA, USA), 900 nM Jpht3, Jpth4 primers, 250 nM P-Jpth2 probe and 8 µL of the DNA sample. The optimal thermocycler conditions retained consisted of 95 °C 10 min, then 40 cycles of two-steps of 94 °C for 30 s and 58 °C for 1 min, followed by 98 °C for 10 min. A temperature ramp of 2.5 °C/s was fixed in all PCR steps. The ddPCR reactions with fewer than 10,000 droplets generated were excluded from the analyses; a result was considered positive if at least two positive droplets were individuated.

The samples of SET1 were run in one biological replicate. The samples of SET 2 were run in three biological replicates (for each plant matrix) (*n* = 3). In particular, there were tested samples from 100 pg to 10 fg/ddPCR for SET1 and samples from 10^3^ cfu/mL to 10 cfu/mL for SET2, respectively. Ten healthy samples from each tested plant species previously assessed by real-time PCR were also tested by ddPCR, QX 200 Droplet Reader, QX 200 Droplet Generator, PX1 PCR Plate Sealer, QuantaSoft 1.7.4.0917 (Biorad, Bio-Rad Laboratories Inc., Hercules, CA, USA).

### 4.7. Samples Homogeneity and Performance Criteria Evaluation

The homogeneity was evaluated for samples of SET1 and SET2 by considering the (SD) of Ct values obtained for each sample by real-time PCR. The replicates were considered sufficiently homogeneous with SD < 1.

The following performance criteria, diagnostic specificity, diagnostic sensitivity, accuracy, repeatability, and analytical sensitivity were evaluated on samples of SET2 according to the EPPO PM 7/98 (4) [13]. In particular, the following target samples were assessed: 3 biological replicates (in 3 repetitions, *n* = 9) from 10^7^ to 10^3^ cfu/mL excluding the samples that did not respect the homogeneity parameters (SD > 1). A total of 45 target samples were assessed. As the non-targets, 20 healthy samples of each of the plant matrices were tested. Some interference was observed on healthy matrices by real-time PCR (high Ct value in duplicates or inconsistencies between the two replicates (i.e., Ct/NA value) whose results are reported in Appendix A.

### 4.8. Statistical Methods

The sample mean ± the standard deviation (SD) was calculated for all data of the real-time PCR and ddPCR. Excel Microsoft 365® was employed as the statistic package. The analysis of qualitative data was performed by the calculation of performance criteria as reported in the PM 7/122 (1). The Bland–Altman test was performed through R version 4.0.2, using the packages “base” and “stats” [26].

## Figures and Tables

**Figure 1 ijms-23-04975-f001:**
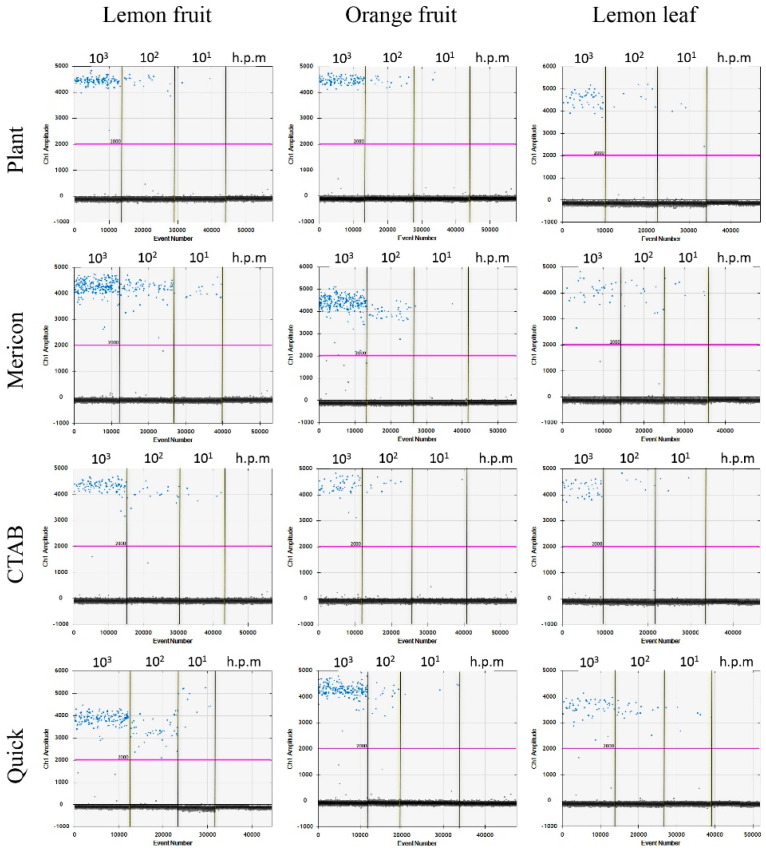
Graphical representation of the results obtained on SET2 samples (from 10^3^ to 10 cfu/mL) by ddPCR. Blue dots indicate positive droplets with amplification while black dots indicate negative dots without amplification. Purple line indicates the threshold separating positive from negative dots; h.p.m. = healthy plant matrix.

**Figure 2 ijms-23-04975-f002:**
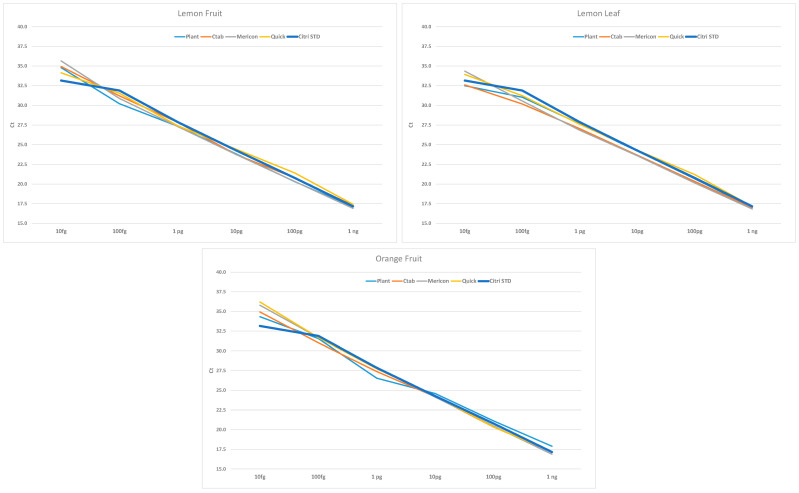
Standard curves represented as linear regression of the quantitation cycle (Ct) values (Y axis) of SET1 samples versus the DNA bacterial concentration of Xcc (X axis); the r^2^ values are reported in Table 1 and Table 3. Different colors indicate the standard curves generated using different DEM (Plant = DNeasy Plant Mini kit, Mericon = DNeasy Mericon Food Kit, Ctab = CTAB extraction method, Quick = QuickPick SML Plant DNA kit) in comparison with the standard curve of Xcc bacterial DNA (Citri STD). DEM = DNA extraction method.

**Figure 3 ijms-23-04975-f003:**
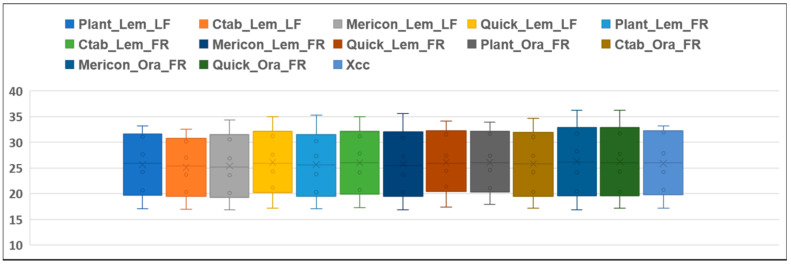
Range of Ct values (*y*-axis) of SET1 considering the different combination of DEM/plant matrices (*x*-axis) in comparison with bacterial DNA. (Plant = DNeasy Plant Mini kit, Mericon = DNeasy Mericon Food Kit, Ctab = CTAB extraction method, Quick = QuickPick SML Plant DNA kit) Lem = lemon, Ora = Orange; LF = leaf; FR = fruit. DEM = DNA extraction method.

**Figure 4 ijms-23-04975-f004:**
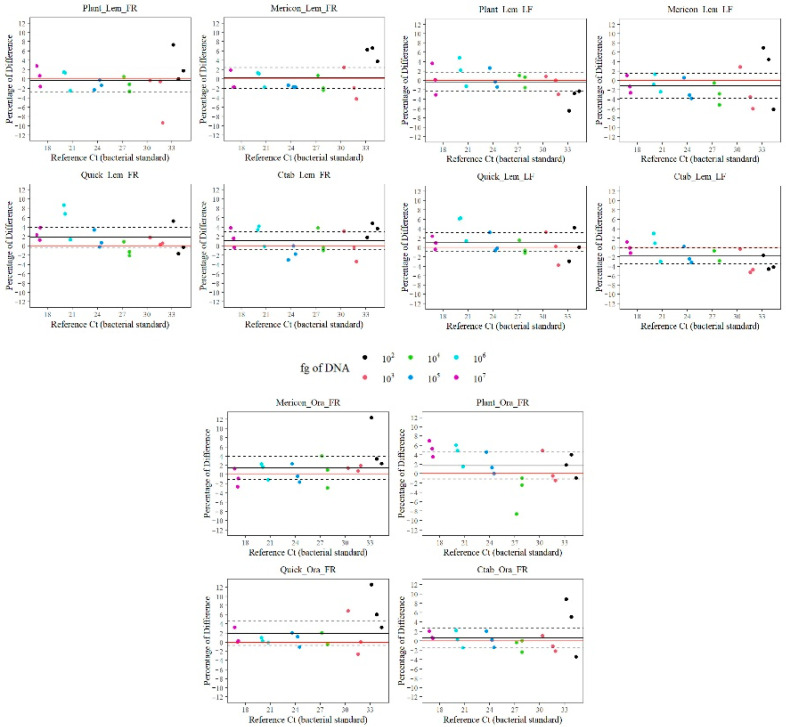
Bland–Altman plot: X axis reports the Ct of Xcc bacterial DNA standard curves, Y axis reports the percentage variation of the DEM with respect to the Xcc bacterial DNA standard. The black horizontal line represents the average (considering all concentrations together) of the percentage variation and indicates whether the method underestimates or overestimates the overall Ct. The red line represents the zero-percentage variation. The black dashed lines define the 95% confidence interval with respect to the black line. DEM = DNA extraction method.

**Table 1 ijms-23-04975-t001:** Results obtained by real-time PCR and droplet digital PCR in samples of SET1. Samples of SET1 were prepared on different plant matrices (lemon leaves and fruits, orange fruit), extracted with the four different DEM (Plant = DNeasy Plant Mini kit, Mericon = DNeasy Mericon Food Kit, CTAB = CTAB-based method, Quick = QuickPick SML Plant DNA kit). For real-time PCR are reported the average of Ct values and the number of positive wells on the total assessed; for ddPCR are reported copies/mL and positive droplets. Grey boxes indicate inconsistent results between the replicates (ΔCt less than 3 cycles between ten-decimal dilutions) and SD > 1. The efficiency, r^2^, and the slope of each real-time PCR are shown. The number of total droplets in ddPCR was >10.000 for all data reported; “-” = not tested samples. DEM = DNA extraction method.

ng of Bacterial DNA per Reaction	Plant														
	Real Time PCR Cubero et al. (2005)									ddPCR Adapted from Cubero et al. (2005)					
	Lemon Fruit			Orange Fruit			Lemon Leaf			Lemon Fruit		Orange Fruit		Lemon Leaf	
	N° pos Wells	Ct Mean	St.dev	N° pos Wells	Ct Mean	St.dev	N° pos Wells	Ct Mean	St.dev	Copies/μL	Positive Droplets	Copies/μL	Positive Droplets	Copies/μL	Positive Droplets
1 ng	3/3	17.10	0.19	3/3	16.27	0.11	3/3	17.03	0.36	-	-	-	-	-	-
100 pg	3/3	20.29	0.08	3/3	19.57	0.05	3/3	20.66	0.22	-	-	-	-	-	-
10 pg	3/3	23.81	0.66	3/3	22.31	0.10	3/3	24.2	0.05	164	1190	342	2407	152	1635
1 pg	3/3	27.33	0.22	3/3	25.82	**1.48**	3/3	27.65	0.33	15.2	157	43	467	14.8	169
100 fg	3/3	30.20	**1.26**	3/3	28.64	0.24	3/3	31.04	0.49	1.1	7	4.7	54	3.2	33
10 fg	3/3	35.30	0.92	3/3	32.99	0.73	3/3	**33.23**	**1.31**	0.42	4	0.53	6	0.34	3
1 fg	3/3	**37.67**	**1.85**	**2/3**	35.92	**1.25**	3/3	36.7	**1.88**	0	0	0.18	2	0.09	1
Std curve parameters	E: 93.60%; r^2^: 0.991; slope: −3.485; Y int: 20.37			E: 103.90%; r^2^: 0.996; slope: −3.232; Y int: 21.21			E: 101.80%; r^2^: 0.994; slope: −3.28; Y int: 20.68								
**ng of Bacterial DNA per Reaction**	**Mericon**														
	**Real time PCR Cubero et al. (2005)**									**ddPCR adapted from Cubero et al. (2005)**					
	**Lemon Fruit**			**Orange Fruit**			**Lemon Leaf**			**Lemon Fruit**		**Orange Fruit**		**Lemon Leaf**	
	**N° pos Wells**	**Ct Mean**	**St.dev**	**N° pos Wells**	**Ct Mean**	**St.dev**	**N° pos Wells**	**Ct Mean**	**St.dev**	**Copies/** **μL**	**Positive Droplets**	**Copies/** **μL**	**Positive Droplets**	**Copies/** **μL**	**Positive Droplets**
1 ng	3/3	16.9	0.14	3/3	16.86	0.19	3/3	16.82	0.11	-	-	-	-	-	-
100 pg	3/3	20.32	0.11	3/3	20.46	0.07	3/3	20.14	0.33	-	-	-	-	-	-
10 pg	3/3	23.74	0.41	3/3	24.14	0.04	3/3	23.6	0.12	60.5	589	138	1073	114	1118
1 pg	3/3	27.30	0.09	3/3	27.58	0.66	3/3	26.83	0.36	11	114	15.1	111	7.1	80
100 fg	3/3	30.88	0.31	3/3	31.7	0.87	3/3	30.55	0.64	1.1	12	1.6	14	0.51	5
10 fg	3/3	35.64	0.40	3/3	36.22	**1.27**	3/3	34.35	**1.82**	0	0	0	**0**	0	**0**
1 fg	3/3	**35.78**	**2.06**	**1/3**	**38.25**	**-**	3/3	**37.7**	**2.74**	0	0	0	**0**	0	**0**
Std curve parameters	E: 99.70%; r^2^: 0.975; slope: −3.328; Y int: 20.47			E: 88.10%; r^2^: 0.997; slope: −3.644; Y int: 20.56			E: 93.20%; r^2^: 0.99; slope: −3.497; Y int: 20.14								
**ng of Bacterial DNA per Reaction**	**Ctab**														
	**Real Time PCR Cubero et al. (2005)**									**ddPCR Adapted from Cubero et al. (2005)**					
	**Lemon Fruit**			**Orange Fruit**			**Lemon Leaf**			**Lemon Fruit**		**Orange Fruit**		**Lemon Leaf**	
	**N° pos Wells**	**Ct Mean**	**St.dev**	**N° pos Wells**	**Ct Mean**	**St.dev**	**N° pos Wells**	**Ct Mean**	**St.dev**	**Copies/** **μL**	**Positive Droplets**	**Copies/** **μL**	**Positive Droplets**	**Copies/** **μL**	**Positive Droplets**
1 ng	3/3	17.27	0.16	3/3	17.17	0.09	3/3	16.98	0.36	-	-	-	-	-	-
100 pg	3/3	20.78	0.17	3/3	20.33	0.16	3/3	20.33	0.22	-	-	-	-	-	-
10 pg	3/3	24.16	0.74	3/3	24.18	0.12	3/3	23.67	0.05	159	1402	145	1522	214	2257
1 pg	3/3	27.84	0.34	3/3	27.38	0.41	3/3	27.03	0.33	17.5	170	19.4	201	27.4	298
100 fg	3/3	31.20	0.33	3/3	31.02	0.30	3/3	30.18	0.49	1	11	1.4	13	1.6	18
10 fg	3/3	34.93	**1.02**	3/3	34.66	**1.54**	3/3	32.59	**1.31**	0.1	**1**	0.11	1	0.54	6
1 fg	**1/3**	**38.44**	**-**	1/3	**38.76**	**-**	3/3	**34.39**	**1.88**	0	**0**	0	**0**	0.38	4
Std curve parameters	E: 92.20%; r^2^: 0.996; slope: −3.525; Y int: 20.75			E: 92.50%; r^2^: 0.994; slope: −3.517; Y int: 20.50			E: 111.4%; r^2^: 0.993; slope: −3.075; Y int: 20.43								
**ng of bacterial DNA per reaction**	**Quick**														
	**Real time PCR Cubero et al. (2005)**									**ddPCR adapted from Cubero et al. (2005)**					
	**Lemon Fruit**			**Orange Fruit**			**Lemon Leaf**			**Lemon Fruit**		**Orange Fruit**		**Lemon Leaf**	
	**N° pos Wells**	**Ct Mean**	**St.dev**	**N° pos Wells**	**Ct Mean**	**St.dev**	**N° pos Wells**	**Ct Mean**	**St.dev**	**Copies/** **μL**	**Positive Droplets**	**Copies/** **μL**	**Positive Droplets**	**Copies/** **μL**	**Positive Droplets**
1 ng	3/3	17.41	0.36	3/3	17.18	0.09	3/3	17.15	0.15	-	-	-	-	-	-
100 pg	3/3	21.40	0.32	3/3	20.35	0.35	3/3	21.2	0.16	-	-	-	-	-	-
10 pg	3/3	24.42	0.22	3/3	24.29	0.24	3/3	24.3	0.17	125	1197	147	1328	127	1163
1 pg	3/3	27.40	0.13	3/3	27.43	0.01	3/3	27.59	0.07	2.1	112	17.8	177	14.4	149
100 fg	3/3	31.54	0.59	3/3	31.68	0.85	3/3	31.23	0.48	0.24	23	1.9	19	2.5	28
10 fg	3/3	34	0.86	3/3	36.21	0.98	3/3	33.94	**1.56**	0.17	3	0	**0**	0.19	2
1 fg	**1/3**	**40.53**	**-**	2/3	**38.31**	0.21	**2/3**	**36.87**	**1.73**	0	2	0	**0**	0	**0**
Std curve parameters	E: 93.50%; r^2^: 0.987; slope: −3.488; Y int: 20.94			E: 89.30%; r^2^: 0.997; slope: −3.607; Y int: 20.65			E: 100.40%; r^2^: 0.99; slope: −3.312; Y int: 20.90								

**Table 2 ijms-23-04975-t002:** Results obtained by real-time PCR and ddPCR in spiked samples of lemon fruit and leaves and orange fruit of SET2 extracted with the four different DEM (Plant = DNeasy Plant Mini kit, Mericon = DNeasy Mericon Food Kit, CTAB = CTAB-based method, Quick = QuickPick SML Plant DNA kit). For real-time PCR are reported the Ct values mean ± SD and the number of positive replicates/number of replicates analyzed; for ddPCR are reported the positive droplets mean, the copies/μL mean ± SD and the number of positive replicates/number of replicates analyzed. Grey boxes indicate inconsistent results between the replicates (ΔCt less than 3 cycles between ten-decimal dilutions) and SD > 1. The number of total droplets in dd-PCR was >10.000 for all data considered for the analyses. Not tested samples are indicate as follows (-). DEM = DNA extraction method.

	Real Time PCR Cubero et al. (2005)									ddPCR Adapted from Cubero et al. (2005)											
	Lemon Fruit			Orange Fruit			Lemon Leaf			Lemon Fruit				Orange Fruit				Lemon Leaf			
Bacterial CFU per Reaction	Plant																				
	N° pos Wells	Ct Mean	St.dev	N° pos Wells	CT MEAN	St.dev	N° pos Wells	Ct Mean	St.dev	N° pos Wells	Positive Droplets Mean	Copies/μL Mean	St.dev	N° pos Wells	Positive droplets Mean	Copies/μL Mean	St.dev	N° pos Wells	Positive Droplets Mean	Copies/μL Mean	St.dev
10^7^	9/9	16.40	0.64	9/9	18.4	0.35	9/9	18.5	0.71	-	-	-	-	-	-	-	-	-	-	-	-
10^6^	9/9	19.93	0.16	9/9	20.3	0.18	9/9	22.9	**2.14**	-	-	-	-	-	-	-	-	-	-	-	-
10^5^	9/9	23.06	0.46	9/9	24.2	1	9/9	25.5	0.66	-	-	-	-	-	-	-	-	-	-	-	-
10^4^	9/9	26.48	0.26	9/9	27.2	0.3	9/9	29.7	1.79	-	-	-	-	-	-	-	-	-	-	-	-
10^3^	9/9	29.77	0.20	9/9	29.5	0.49	9/9	31.9	0.94	3/3	128	10.3	1.15	3/3	149	12.8	2.51	3/3	20.7	2.3	2.51
10^2^	9/9	32.96	0.48	**8/9**	33.2	**1.15**	9/9	34.1	0.56	3/3	16.3	1.22	0.68	3/3	13.7	1.18	0.90	3/3	4.66	0.45	0.32
10^1^	**7/9**	36.10	**1.43**	9/9	34.8	**1.34**	**5/9**	36.52	**2.04**	**1/3**	**1**	0.08	0.08	2/3	5.66	0.45	0.64	2/3	2.66	0.19	0.12
Bacterial CFU per reaction	Mericon																				
	N° pos wells	Ct mean	St.dev	N° pos wells	Ct mean	St.dev	N° pos wells	Ct mean	St.dev	N° pos wells	Positive droplets mean	Copies/μL mean	St.dev	N° pos wells	Positive droplets mean	Copies/μL mean	St.dev	N° pos wells	Positive droplets mean	Copies/μL mean	St.dev
10^7^	9/9	17.29	0.16	9/9	19.3	0.19	9/9	17.5	0.78	-	-	-	-	-	-	-	-	-	-	-	-
10^6^	9/9	19.92	0.29	9/9	21.7	0.8	9/9	20.9	0.33	-	-	-	-	-	-	-	-	-	-	-	-
10^5^	9/9	22.56	0.35	9/9	24.7	0.28	9/9	24	0.2	-	-	-	-	-	-	-	-	-	-	-	-
10^4^	9/9	26.09	0.36	9/9	27.9	0.51	9/9	28.2	0.8	-	-	-	-	-	-	-	-	-	-	-	-
10^3^	9/9	28.95	0.41	9/9	30.2	0.32	9/9	31.6	0.74	2/3	177	11.4	6.57	3/3	206	18.9	2.90	3/3	22	2.13	0.76
10^2^	9/9	32.08	**1.24**	9/9	33.1	**1.7**	9/9	33.3	**2.35**	3/3	33	2.75	2.92	3/3	22.7	2.06	1.19	2/3	5.5	0.53	0.24
10^1^	9/9	**32.3**	**1.69**	**7/9**	36.1	**1.14**	**7/9**	**34.5**	**1.58**	3/3	11.7	1.1	0.34	2/3	**1.66**	0.13	0.05	**1/3**	3.66	0.36	0.63
Bacterial CFU per reaction	Ctab																				
	N° pos wells	Ct mean	St.dev	N° pos wells	Ct mean	St.dev	N°pos wells	Ct mean	St.dev	N° pos wells	Positive droplets mean	Copies/μL mean	St.dev	N° pos wells	Positive droplets mean	Copies/μL mean	St.dev	N° pos wells	Positive droplets mean	Copies/μL mean	St.dev
10^7^	9/9	16.53	0.56	9/9	20.1	0.23	9/9	18.3	0.36	-	-	-	-	-	-	-	-	-	-	-	-
10^6^	9/9	20.79	0.17	9/9	22.2	0.74	9/9	22.3	0.16	-	-	-	-	-	-	-	-	-	-	-	-
10^5^	9/9	23.850	0.52	9/9	25.1	0.22	9/9	25.7	0.19	-	-	-	-	-	-	-	-	-	-	-	-
10^4^	9/9	28.75	**2.09**	9/9	29.2	0.99	9/9	28.8	0.61	-	-	-	-	-	-	-	-	-	-	-	-
10^3^	9/9	30.24	0.45	9/9	30.1	0.66	9/9	31.6	0.67	2/3	103	8	0.28	3/3	79.7	6.9	2.38	3/3	46	4.13	2.22
10^2^	9/9	32.7	**1.04**	9/9	34.3	**1.09**	9/9	33.5	0.78	2/3	23.5	1.32	1.94	2/3	6	0.57	0.41	3/3	10.7	1.03	0.92
10^1^	9/9	**32.9**	**1.72**	**7/9**	37.6	**2.31**	**9/9**	**34.6**	**1.9**	2/3	9.5	0.75	1.06	**1/3**	**1**	0.09	0.09	2/3	3.5	1.51	0.10
Bacterial CFU per Reaction	Quick																				
	N° pos wells	Ct mean	St.dev	N° pos wells	Ct mean	St.dev	N°pos wells	Ct mean	St.dev	N° pos wells	Positive droplets mean	Copies/μL mean	St.dev	N° pos wells	Positive droplets mean	Copies/μL mean	St.dev	N° pos wells	Positive droplets mean	Copies/μL mean	St.dev
10^7^	9/9	17.3	0.16	9/9	18.8	0.17	9/9	17.3	0.69	-	-	-	-	-	-	-	-	-	-	-	-
10^6^	9/9	19.6	0.8	9/9	20.9	0.33	9/9	19.4	0.36	-	-	-	-	-	-	-	-	-	-	-	-
10^5^	9/9	22.4	0.78	9/9	24.2	0.44	9/9	23	0.17	-	-	-	-	-	-	-	-	-	-	-	-
10^4^	9/9	25.4	0.15	9/9	27.2	0.37	9/9	26.9	0.46	-	-	-	-	-	-	-	-	-	-	-	-
10^3^	9/9	29.8	0.74	9/9	29.4	0.34	9/9	30.8	0.66	2/3	99	8.9	7.35	3/3	177	17.5	2.46	3/3	50.3	4.46	1.70
10^2^	9/9	**30.5**	**1.47**	9/9	32.8	0.51	9/9	32.8	0.53	2/3	67.6	5.36	7.23	2/3	16	1.55	0.49	2/3	15	1.35	0.49
10^1^	**8/9**	**32.1**	**1.72**	**6/9**	35.20	0.97	**9/9**	**35.3**	**1.18**	3/3	16.7	1.74	2.65	2/3	3.5	0.3	0.04	3/3	11	0.82	0.68

**Table 3 ijms-23-04975-t003:** Results obtained by real-time PCR and droplet digital PCR in samples of ten-fold dilution of bacterial DNA from strain Xcc NCPPB 3234, pathotype A were prepared as a standard reference curve. For real-time PCR are reported the average of Ct values and the number of positive wells on the total assessed; for ddPCR are reported copies/mL and positive droplets. Grey boxes indicate inconsistent results between the replicates (ΔCt less than 3 cycles between ten-decimal dilutions) and SD > 1. The efficiency, r^2^, and the slope of each real-time PCR are shown. The number of total droplets in ddPCR was >10.000 for all data reported; “-” = not tested samples.

Bacterial Culture (NCPPB 3234)					
Real Time PCR Cubero et al. (2005)				ddPCR Adapted from Cubero et al. (2005)	
ng per Reaction	N° pos Wells	Ct Mean	St.dev	Copies/µL	Positive Droplets
1ng	3/3	17.15	0.23	-	-
100pg	3/3	20.77	0.44	-	-
10pg	3/3	24.26	0.47	163	1971
1pg	3/3	27.86	0.39	13.30	135
100fg	3/3	31.88	0.81	1.80	22
10fg	3/3	**33.16**	0.60	0.16	2
1fg	3/3	37.77	0.42	0.09	**1**
Std curve parameters	E: 98.7%; r^2^: 0.984; slope: −3.353; Y int: 21.845				

**Table 4 ijms-23-04975-t004:** Percentage of healthy samples which yielded high Ct in duplicates or in one out of two replicates (i.e., Ct value/NA) for each DEM/plant matrices combinations (Plant = DNeasy Plant Mini kit; Mericon = DNeasy Mericon Food Kit, Ctab = CTAB extraction method, Quick = QuickPick SML Plant DNA kit). The raw data are reported in Appendix A. DEM = DNA extraction method.

	Mericon	Ctab	Quick	Plant
**Lemon Fruit**	30%	50%	5%	20%
**Orange Fruit**	20%	40%	25%	10%
**Lemon Leaf**	35%	40%	5%	45%
**Mean**	28%	43%	11%	25%

**Table 5 ijms-23-04975-t005:** Values of performance criteria obtained by real-time PCR diagnostic sensitivity, diagnostic specificity and accuracy. Positive agreement (PA), negative agreement (NA), positive deviation (PD) and negative deviation (ND) obtained by testing spiked samples contaminated from 10^7^–10^3^ cfu/mL of Xcc SET2 and non-spiked healthy samples.

Performance Criteria	Lemon Fruit				Orange Fruit				Lemon Leaves			
	Plant	Mericon	Ctab	Quick	Plant	Mericon	Ctab	Quick	Plant	Mericon	Ctab	Quick
**PA**	45	45	45	45	45	45	45	45	45	45	45	45
**ND**	0	0	0	0	0	0	0	0	0	0	0	0
**NA**	20	20	20	20	20	20	20	20	20	20	20	20
**PD**	0	0	0	0	0	0	0	0	0	0	0	0
**Target**	45	45	45	45	45	45	45	45	45	45	45	45
**Non-target**	20	20	20	20	20	20	20	20	20	20	20	20
**Total samples**	65	65	65	65	65	65	65	65	65	65	65	65
**Diagnostic sensitivity**	100%	100%	100%	100%	100%	100%	100%	100%	100%	100%	100%	100%
**Diagnostic specificity**	100%	100%	100%	100%	100%	100%	100%	100%	100%	100%	100%	100%
**Accuracy**	100%	100%	100%	100%	100%	100%	100%	100%	100%	100%	100%	100%

**Table 6 ijms-23-04975-t006:** Analytical sensitivity (ASE) obtained by using different DNA extraction method (DEM) on spiked samples (SET2).

		Plant Matrices		
DEM	Detection Method	Lemon Fruit	Orange Fruit	Lemon Leaf
**Plant**	**real-time PCR**	10^2^	10^3^	10^2^
**Mericon**		10^3^	10^3^	10^3^
**Ctab**		10^3^	10^3^	10^2^
**Quick**		10^3^	10^2^	10^2^
**Plant**	**ddPCR**	10^2^	10	10
**Mericon**		10	10^2^	10
**Ctab**		10	10^2^	10
**Quick**		10	10	10

**Table 7 ijms-23-04975-t007:** Plant species, matrix, place and year of sampling.

Plant Specie	Matrix	Place of Sampling	Year of Sampling
Orange	Fruit	Rome (Italy)	2020
Lemon	Leaves	Latina (Italy)	2020
Lemon	Fruit	Naples (Italy)	2020

## Data Availability

Not applicable.

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
