# Peer review of "Intra-Laboratory Evaluation of DNA Extraction Methods and Assessment of a Droplet Digital PCR for the Detection of Xanthomonas citri pv. citri on Different Citrus Species"

_ijms, 2022, doi:10.3390/ijms23094975_

Round 1
Reviewer 1 Report
Title manuscript: Intralaboratory evaluation of DNA extraction methods and 2 assessment of a droplet digital PCR for the detection of 3 Xanthomonas citri pv. citri on different Citrus species
In this work, the aim of authors is to validate DNA extraction methods linked to the Xanthomonas citri pv. citri bacteria detection. At this regard the qPCR diagnostic method was compared with ddpCR technology. The work is interesting and can be useful for improve the set-up pathogen detection protocol.
For this aim, the authors analysed several dilutions of the bacterium or DNA from bacteria spiked with DNA from different host plants specie and matrix or the plant extracts from the latters. However, the description of the procedure is not entirely clear. Furthermore, the results should suggest clearly a limit of detection (LOD) and quantification (LOQ) according to the tested procedures for both Real Time quantitative PCR (qPCR) and droplet digital PCR (ddPCR). This involves suitable numbers of technical replicas for each thesis, which is not entirely evident in the work. Various inaccuracies and omissions are included in the manuscript about the data shown in the tables. See below for details
Line 22: change qPCR standard curves with quantitative (q) PCR
Line 84: ‘SET1 and SET2’
Briefly indicate what they refer to
Line 89: ‘Ct values’
Indicate the first time the Ct as threshold cycle. In addition, about the correct nomenclature, the Cq instead of Ct are more appropriate according to the MIQE procedure for Real Time PCR technology (Bustin et al., 2009)
In Tables 1A and Table 2 change comma ‘,’ with point ‘.’ for decimal number indication
In in the first column on the left of Tables 1A and 2 indicate that they are bacteria
Table 1A, the 1ng and 100 pg (bacterial concentration?) not was tested on ddPCR? Specify better, it is not clear if these concentrations have not been analyzed or if they are saturated/inhibited samples.
Make it clear – also in table 2
The data related to ddPCR are reported without standard deviations? Didn't you do any technical replicas? yet in the text you indicate them. Clarify this and include any necessary replies
Line 131: ‘(at 103 cfu/mL) ‘
What does it mean? That's not what I see in the figure 2
Line 144: ’ E values slightly deviates from the acceptable values ‘
What o mean? Indicated the acceptable value according to the correct reference for qPCR data
Line 149: ‘This was predictable at low target concentrations due to the greater variability between technical replicates’
The limit of detection and quantification could be defined according MIQE procedure as reported in the Bustin et al., 2009:.
‘…Typically, sensitivity is expressed as the limit of detection (LOD), which is the concentration that can be detected with reasonable certainty (95% probability is commonly used) with a given analytical procedure. The most sensitive LOD theoretically possible is 3 copies per PCR (28 ), assuming a Poisson distribution, a 95% chance of including at least 1 copy in the PCR, and single-copy detection.
The type of experiment planned seems to want to investigate the limit of detection (LOD) and quantification (LOQ) for the pathogen detection accordin to different DEM. However, it is never clearly stated. You only indicate 'inconsistent results', it must be indicated more clearly whether it is a limit of detection or quantification. Perhaps not enough technical replies have been made to indicate this? But the validity of your experimental proof passes through this aspect. How do you can indicate one protocol is better than another if you don't clearly state its limits?
Lines 180-183: about the results you report: ‘The results showed that at lowest DNA target con centration (10 fg/ddPCR) orange fruit matrix resulted in the highest number of negative results. However, among DEM, Plant showed reliable results with all plant matrices as shown in Table 1(A). QuickPick showed accurate results with the exception for orange fruit, Ctab was reliable only in samples of lemon leaf and Mericon in none (Table 1A).’
In addition you can add some comments regarding the absence of results reported for1ng and 100 pg baterial concentrations (see above), here it seems clear that the DEM indicated as Mericon shows results decidedly different from the other DEM. Furthermore, I would try to indicate which is the LOD linked to the adopted protocol. Clearly, you can indicate this value with adequate number of technical replicates also for ddPCR. Without technical replicate you can’t indicate any.
Line 194: Table 3.
In the caption was reported: ‘Percentage of healthy samples which yielded high Ct or inconsistencies between the two replicates’
Not unit of measurement in the table are indicated. In addition, percentage between two replicas? Make it clearer both in the table and in the caption
Table 4:
In the caption: ‘Table 4. Values of performance criteria obtained by qPCR [10]: diagnostic sensitivity, diagnostic specificity and accuracy. Positive agreement (PA), negative agreement (NA), positive deviation (PD) and negative deviation (ND) obtained by testing spiked samples contaminated from 107 -103 cfu/mL of Xcc SET2 (A) and non-spiked healthy samples.’
Like table 3, also in the table 4 not unit of measurement in the table are indicated. What does it mean: testing spiked samples contaminated from 107 -103 cfu/mL …..? I don't understand this from the table
Line 346: ‘The strain NCPPB 3234’
Modify with NCPPB 3234 (Xcc),
Line 353: move ‘strains NCPPB 3234’
Line 378: DNA of plant extract’
Report here by what method this DNA was extracted
Line 383: ‘Three independent biological replicates were prepared for each bacterial concentration’
Three replicate for each plant matrix?
Line 390: DNA Extraction
This part should be better indicated, In ‘Sample preparation paragraph was reported: ‘Samples prepared for the intra-laboratory study were grouped in SET1 and SET2. SET1 was prepared by amending DNA of plant extract with ten-fold dilution of Xcc DNA (from 1 ng to 1 fg/qPCR or ddPCR reaction). SET2 consisted of spiked samples prepared by mixing bacterial suspensions at known concentration (CFU/ml) with healthy plant ex- tracts, followed by DEM.’
So, I understand that in set1 you first did the DNA extraction of every single plant matrix and bacteria and then you mixed the samples. In set 2 you mixed bacteria and plant extracts first and then you did the DNA extraction of mixed samples. It’s, correct? Here it is not clear. In this paragraph I understood that bacteria and plant material DNA extraction is done separately also in set2? what's different between set1 and set2?
also, what is the plant DNA concentration in set 1?
Line 406: ‘PAC, NIC and NAC’
Specify these acronyms
Line 406: ‘Samples of SET1 were run in triplicates (n=3) and each of the three biological replicates of SET2 were run in technical triplicate (n = 9)’
Specify better, three biological replicates for each plant extract matrix?
It is not clear how many technical replicas have been made. For each thesis tested that corresponds to each plant matrix + bacteria of each biological replica, how many technical replicas? This is very important for checking the detection limit
Line 435: In particular, the following target samples were assessed: 3 biological replicates (in three repetitions, n=9)
Nine technical replicas for each plant matrix?
Reviewer 2 Report
Citrus bacterial canker ( Xanthomonas citri pv. citri) is an important disease of major economic significance.
I believe that the manuscript is well written. The study done by the team is valuable and deserves to be published in IJMS. I found some inaccuracies that need to be corrected.
- Fig. 2 shows linear regression? Please provide the R value?
- Table 3 Is giving the mean value necessary? In my opinion it can be removed
- Table 4. The caption is under the table. It should be corrected.
- Were statistical methods used in the research? Please describe them.
